# A quality improvement study on the relationship between intranasal povidone-iodine and anesthesia and the nasal microbiota of surgery patients

Eric N. Hammond[1,2], Ashley E. Kates[2,3], Nathan Putman-Buehler[4], Lauren Watson[5], Jared J. Godfrey[2,3], Nicole Brys[6], Courtney Deblois[7,8], Andrew J. Steinberger[7,8], Madison S. Cox[7,8], Joseph H. Skarlupka[7,8], Ambar Haleem[2], Michael L. Bentz[9], Garret Suen[7], Nasia Safdar[2,3] *

1 Institute for Clinical and Translational Research, University of Wisconsin-Madison, Madison, WI, United States of America, 2 Division of Infectious Disease, Department of Medicine, University of Wisconsin School of Medicine and Public Health, Madison, WI, United States of America, 3 William S. Middleton Memorial Veterans Hospital, Madison, WI, United States of America, 4 Department of Biochemistry, College of Agricultural and Life Sciences, University of Wisconsin-Madison, Madison, WI, United States of America, 5 SSM Health, St. Mary's Hospital, Madison, WI, United States of America, 6 Waisman Center, University of Wisconsin-Madison, Madison, WI, United States of America, 7 Department of Bacteriology, University of Wisconsin-Madison, Madison, WI, United States of America, 8 Microbiology Doctoral Training Program, University of Wisconsin-Madison, Madison, WI, United States of America, 9 Division of Plastic and Reconstructive Surgery and Urology, Department of Surgery, University of Wisconsin School of Medicine and Public Health, Madison, WI, United States of America

☯ These authors contributed equally to this work.
* ns2@medicine.wisc.edu

**Data Availability Statement:** All sequences associated with this study were deposited into the National Center for Biotechnological Information's

## Abstract

### Introduction

The composition of the nasal microbiota in surgical patients in the context of general anesthesia and nasal povidone-iodine decolonization is unknown. The purpose of this quality improvement study was to determine: (i) if general anesthesia is associated with changes in the nasal microbiota of surgery patients and (ii) if preoperative intranasal povidone-iodine decolonization is associated with changes in the nasal microbiota of surgery patients.

### Materials and methods

One hundred and fifty-one ambulatory patients presenting for surgery were enrolled in a quality improvement study by convenience sampling. Pre- and post-surgery nasal samples were collected from patients in the no intranasal decolonization group (control group, n = 54). Pre-decolonization nasal samples were collected from the preoperative intranasal povidone-iodine decolonization group (povidone-iodine group, n = 97). Intranasal povidone-iodine was administered immediately prior to surgery and continued for 20 minutes before patients proceeded for surgery. Post-nasal samples were then collected. General anesthesia was administered to both groups. DNA from the samples was extracted for 16S rRNA sequencing on an Illumina MiSeq.

Short Read Archive (NCBI SRA database) and are available under BioProject Accession PRJNA859807.

**Funding:** The author(s) received no specific funding for this work.

**Competing interests:** The authors have declared that no competing interests exist.

## Results

In the control group, there was no evidence of change in bacterial diversity between pre- and post-surgery samples. In the povidone-iodine group, nasal bacterial diversity was greater in post-surgery, relative to pre-surgery (Shannon's Diversity Index (P = 0.038), Chao's richness estimate (P = 0.02) and Inverse Simpson index (P = 0.027). Among all the genera, only the relative abundance of the genus *Staphylococcus* trended towards a decrease in patients after application (FDR adjusted P = 0.06). Abundant genera common to both povidone-iodine and control groups included *Staphylococcus*, *Bradyrhizobium*, *Corynebacterium*, *Dolosigranulum*, *Lactobacillus*, and *Moraxella*.

## Conclusions

We found general anesthesia was not associated with changes in the nasal microbiota. Povidone-iodine treatment was associated with nasal microbial diversity and decreased abundance of *Staphylococcus*. Future studies should examine the nasal microbiota structure and function longitudinally in surgical patients receiving intranasal povidone-iodine.

## Introduction

The human nasal microbiota structure is complex. It comprises diverse microbial communities [1] that play a vital role in the health of an individual [2,3]. Commensal bacteria (*Corynebacterium spp*. and *Staphylococcus epidermidis*) help to control and maintain nasal microbial community diversity [4,5]. Despite the presence of these commensals, pathogenic organisms such as *Staphylococcus aureus* can evade the nasal immune system and reside in the nares for a long time without causing disease to the host. However, when the healthy nasal microbiota is disrupted, such as during surgery, each member of the microbial community competes for adhesion sites, space, and nutrients [2,3,6]. This results in rapid multiplication of opportunistic pathogens, allowing them to readily spread from the nares to other body sites and cause infections [2,3,7] in addition to surgical site infections (SSIs) [7].

According to the Centers for Disease Control and Prevention's (CDC) definition, SSI is an infection that occurs after surgery in the part of the body where the surgery took place [8]. The most prevalent pathogen associated with nasal SSI is *S. aureus* [9–11]. Genotypic studies have shown up to 80% of *S. aureus* infections are caused by a patient's own nasal microbiota [12]. Across the United States between 20% and 40% of patients have nasal carriage of *S. aureus* [13] increasing their risk of SSI [8,14].

In healthcare settings, *S. aureus* colonization of surgical wounds can occur through the patient's own nasal microbiota [12], through the hands of a healthcare worker, from patient to patient, and from contaminated healthcare environments [15]. *S. aureus* can cause invasive infections such as bloodstream infection and sepsis [9,16,17]. In addition, *S. aureus* can form biofilms which impede and complicate wound healing [18], rendering *S. aureus* infections extremely difficult to treat.

Given these dangers, and the World Health Organization's recommendations [14], many surgical units have recently implemented multiple strategies directed at *S. aureus* to prevent SSIs. For instance, prior to surgery, patients are recommended to decolonize their nares at least twice per day for 5 days with Mupirocin ointment before surgery. On the day of surgery, patients are also administered with an antibiotic prophylaxis complemented with

chlorhexidine gluconate (CHG) body bath or wipe, and antiseptic mouth rinsing with Hibi-clens (CHG 0.12%) or povidone-iodine (PI) (1%) [19].

Consistent adherence to 5 days of nasal Mupirocin application can be challenging for patients [20] and there have been reports of Mupirocin resistance [21–23]. Thus, there is a need to investigate the effectiveness of an alternative intranasal non-antibiotic-based decoloni-zation agent, such as PI, to prevent SSIs.

PI has a broad antibacterial spectrum [24], minimal adverse effects, is easy to use, and can be applied within an hour prior to surgery by a surgical nurse [25]. Additionally, studies have shown that PI can prevent and disrupt microbial biofilms [26,27]. This suggests that PI admin-istered nasally may help to reduce SSIs. As a result, many hospitals in the United States, have introduced the use of intranasal PI as part of standard preoperative protocol. The general oper-ative protocol involves general anesthesia to enhance the procedure. Here, it is unclear what impact anesthesia alone has on the nasal microbiota.

To date, there is no study evaluating the effect of anesthesia on the nasal microbiota. Recent studies on PI are predominately based on the effect of PI on SSI rates [28–30]; however, there is limited research on the impact of PI on the nasal microbiota of surgical patients. In this qual-ity improvement (QI) study, we sought to determine: (i) if general anesthesia can modulate the nasal microbiota and (ii) if PI decolonization is associated with changes in the nasal microbiota.

## Materials and methods

### Ethical approval

We conducted this quality improvement (QI) study on surgical patients due for surgery at the ambulatory surgery center (an academic medical center), located in the midwestern United States. We applied the quasi-experimental study design. The study was designed to improve presurgical treatment in the academic medical center and specimens were not connected to any patient identifiers. The study protocol was reviewed and approved by the academic medi-cal center quality improvement committee. The University of Wisconsin Health Sciences Insti-tutional Review Board exempted this study as a quality improvement project (approval number 2019–0466, May 6, 2019) and the nurses obtained verbal consent from each patient.

### Patient populations

The nasal samples were collected between May—July 2019 and January—February 2020. The patients were divided into two groups (PI group and general anesthesia only group). All patients received general anesthesia and antibiotic prophylaxis during surgery. Patients were administered with intravenous antimicrobial prophylaxis: cefazolin or clindamycin if allergic to the former. Patients with known Methicillin-resistant *Staphylococcus aureus* colonization had vancomycin 15mg/kg in addition to cefazolin [25]. The PI group were those who received intranasal PI decolonization which lasted for 20 minutes before proceeding for surgery. The general anesthesia only group, were those who received only general anesthesia without PI or any other nasal decolonizing agents prior to surgery. The general anesthesia only group also served as controls and hereinafter will be referred to as the control group. These patients received anesthetic gases inhaled through nose or mouth with a mask or breathing tube sup-plemented with propofol, which was administered intravenously in the arm. The type of anes-thesia administered was selected on a case-by-case basis by the anesthesiologist.

We used an open-label strategy in which patients, surgical nurses, and researchers knew the type of nasal decolonization agent assigned to a given patient. We used convenience sampling to select and enroll patients as they reported to the preoperative room for surgery. In the

preoperative room, the surgical nurses explained the relevance of PI decolonization, potential side effects, and nasal sample collection process to the patients. Once a patient agreed to participate and provided verbal consent, they were enrolled into the study [25]. Patients included those scheduled for cosmetic plastic surgery–breast reduction, breast augmentation, and tummy tuck. Patients scheduled for a major nasal surgery were excluded. In the PI group, patients with known allergies to iodine were also excluded. Patients without general anesthesia and those who refused to provide two nasal samples (pre- and post-surgery) were also excluded from the study.

## Pre- and post-surgery nasal sample collection

All pre- and post-surgical nurses working in the academic medical center were trained in nasal sample collection, documentation, labelling, and sample storage. To obtain nasal samples, trained surgical nurses gently inserted a dry dual-headed BD BBL™ CultureSwab™ Sterile, Media-free Swab (Becton, Dickinson and Company (BD), Sparks, Maryland, USA) tip into the widest part (about 3–5 mm) of the anterior nares of the patient and gently spun 4 times with slight pressure to collect the sample. Swabs were immediately placed in transport tubes pre-labelled with a unique study identifier and stored at 4°C until laboratory processing. The first time timepoint (pre-) occurred immediately prior to application of the nasal PI solution (before the patient underwent surgery). The second time point (post-) occurred after surgery but before the patient was discharged. There was no specific time allocated to post-surgery nasal sample collection.

## Sample storage and transportation

Samples were transported at 4°C in a cooler from the academic medical center to the University of Wisconsin-Madison Infection Disease Research Laboratory (IDR), where they were immediately stored at 4°C. The IDR Lab received all samples within 48 hours of collection. Each sample was processed by cutting off each head with sterilized scissors into a sterile 2.0 mL bead beating tube containing 500 mg of 0.1 mm diameter zirconia/silica beads (Biospec Products, Oklahoma, USA) aseptically and stored at -80°C until DNA extraction and sequencing.

## DNA extraction, amplification, and sequencing

We followed a previously published protocol [31] with some modifications to extract DNA. Briefly, each nasal swab specimen was exposed to enzymatic lysis and mechanical lysis using silica beads. Extract was further purified and quantified on BioTek Synergy 2 plate reader (BioTek, Winooski, VT, USA) using Qubit fluorometric quantitation reagents (Thermo Fisher Scientific, Waltham, MA, USA).

The fourth hypervariable (V4) region of the bacterial 16S rRNA gene was amplified using the one-step polymerase chain reaction (PCR) approach with barcoded V4 primers (F- GTGCCAGCMGCCGCGGTAA; R- GGACTACHVGGGTWTCTAAT). Each of these primers were barcoded with individual custom indices to facilitate demultiplexing, as described in Kozich et al. (2013) [32]. Each PCR reaction consisted of 12.5 μl KAPA 2x HiFi Master Mix (KAPA Biosystems, Wilmington, MA, USA), 0.5 μl of 10 μM forward primer, 0.5 μl of 10 μM reverse primer and up to 11.5 μl of 10ng/μl DNA to a total volume of 25 μl with nuclease-free water (IDT, Coralville, Iowa, USA). Amplification conditions on a *C1000 Touch™ thermal cycler* (Bio-Rad Laboratories, Hercules, CA, USA) were 95°C for 3 min, 35 cycles of 95° for 30 s, 55°C for 30 s, and 72°C for 30 s, followed by a final extension at 72°C for 5 min. The PCR products were purified by running on a 1% low-melt agarose gel stained with SYBR Safe DNA

Gel Stain (Invitrogen, Waltham, CA) to isolate amplicons of the expected size. DNA bands of ~380 bp were excised and purified with the Zymo Gel DNA Recovery Kit (Zymo Research, Irvine, CA, United States). Purified PCR products were equimolar pooled, then sequenced on an Illumina MiSeq (Illumina, San Diego, CA, USA) with 10% PhiX control using a 500-cycle v2 sequencing kit and custom sequencing primers [32].

**Quality control.** For quality control purposes negative and positive controls were sequenced alongside with patients' samples to validate: (i) DNA isolation, (ii) DNA purification, and (iii) sequencing and data analysis. We used positive controls, ZymoBIOMICS® Gut Microbiome Standard (Zymo Research Corp., Irvine, CA) and negative controls (non-sample blanks).

## Microbiota analysis

**Sequence data clean-up in mothur.** The raw sequences were demultiplexed according to their sample-specific indices on the Illumina MiSeq. We subsequently cleaned-up the data in the program mothur (v1.44.2) [32]. Paired end reads were computationally stitched together to form contigs and aligned against the SILVA 16S rRNA gene reference alignment database, release 138 [33,34]. Contigs that were not aligned to the V4 region were eliminated from analysis. We performed pre-clustering to reduce sequencing errors with a defect value of 2. The *UCHIME* algorithm [35] in mothur was used to identify and removed chimeric sequences. Singletons were removed from the dataset before performing operational taxonomic unit (OTU) clustering.

The remaining high-quality sequences were clustered into OTUs with 97% similarity threshold using the cluster.split (method = opti) in mothur [36]. Bacterial sequences were classified against the SILVA database (v. 138) with a bootstrap cutoff of 80. Sequence coverage was calculated in mothur with Good's coverage index [37] to assess if enough sequencing was done to have an accurate picture of true sample diversity. Samples with a pre-normalization Good's coverage of less than 77% were excluded from the analysis.

After cleanup in mothur, we predicted and removed suspected contaminants with the decontam [38] package in R (version 3.6.1) [39] using the prevalence method [40]. We also removed all quality control samples prior to normalization. Finally, we normalized our raw reads count to 400 reads/sample in mothur, which represented the lowest number of sequences that ensured sufficient coverage for all samples. This is to account for differences in sequencing depth between samples prior to statistical analysis [41]. Alpha diversity metrics were calculated in mothur.

## Statistical analysis

Statistical analyses were performed in R (version 3.6.1) [39] using RStudio v1.2.1335 [42]. Any patient without both pre- and post-samples after normalization were removed to allow for paired analysis. The Wilcoxon signed rank test was applied to determine differences in alpha diversity (within-sample-diversity) metrics between paired pre- and post-surgery samples in PI group and the control group separately. We also used Wilcoxon rank-sum test to compare changes between PI-post and control-post. For the exploratory analysis, we randomly subsampled PI group to equal the number of samples in the control group (n = 46 per group) to determine changes in alpha diversity. Beta diversity (between-sample diversity) was assessed using the Bray-Curtis dissimilarity metric [43] and visualized using non-metric multidimensional scaling (nMDS, vegan::metaMDS) plots of square root transformed data. We further added the standard error ellipses around the plotted points to better illustrate the behavior of pre and post centroid points within groups. The Beta dispersion test (vegan::betadisper) was used to

determine equality of group variance and the analysis to perform. We used PERMANOVA [44] (vegan::adonis2) to test if beta diversity centroids differed among the pre and post samples in both control and PI groups. We then identified OTUs that abundantly contributed to the differences observed between pre- and post- for each treatment using the similarity or percentages (SIMPER) function (vegan::simper). OTUs that contributed to more than 1% of the differences between groups were subject to Kruskal-Wallis rank sum tests with false discovery rate (fdr) *P*-value correction to confirm differential abundance and control for false positives. An exploratory analysis was additionally performed to directly compare the changes in *Staphylococcus* relative abundances between control and PI groups.

## Results

A total of 151 surgical patients were involved in our quality improvement (QI) study. Patients were divided into two groups: a PI group (97 patients with 194 nasal samples collected) and a control group (54 patients with 108 nasal samples collected). In the PI group, PI stayed in patients' nares throughout the duration of surgery. In the PI group, the average time between nasal swabs was 3 hours (range:1.15–9 hours). Duration of PI was determined by the projected length of the surgery. In the control group, the average time between nasal swabs was 3.5 hours (range:1.25–6 hours). A study schematic diagram outlines various components in Fig 1.

In total, 3,967,385 raw sequences were generated from the 357 samples. Following decontamination and clean up in mothur, 2,990,013 sequences remained (mean = 8,398.91, SD = 14,638.16) resulting in 14,200 OTUs. Post-normalization, a total of 130 patients with 260 sequenced nasal samples representing 2,867,890 bacteria sequences were retained in the final dataset for analysis. This comprised of 84 PI patients with 168 samples (1,239,198 sequences, mean = 7,376.179, SD = 9,532.848) and 46 control patients with 92 samples (1,628,692 sequences, mean = 1,7703.17, SD = 2,4492.52). We excluded 42 samples (PI = 26, control = 16) from analysis due to insufficient DNA yield, failure to pass normalization, and lack of paired samples. All positive and negative control samples were also excluded from the final data for analysis. A summary of sample collection results can be found in S1–S3 Tables.

### Response of nasal bacterial diversity to anesthesia

In the control group, we observed diverse microbial communities with varied bacterial community compositions (Fig 2A and 2C and S4 Table). Some of the phyla present included Actinobacteriota, Firmicutes, Proteobacteria, and Bacteriodota. The genera detected included *Staphylococcus*, *Pseudomonas*, *Corynebacterium* and *Lactobacillus*. See S4 Table for relative abundance data.

We assessed alpha diversity (within-sample diversity) of patient nasal community samples pre- and post-surgery. In the control group there was no change in diversity between pre- and post-surgery patients with respect to Chao's richness (P = 0.739), Shannon's diversity index (P = 0.241) or Inverse Simpson index (P = 0.122) Fig 3A, 3C and 3E). The Bray-Curtis dissimilarity showed that the overall bacterial community composition between pre- and post-control samples were not different from each other (P = 0.974, PERMANOVA, nMDS stress = 0.25). A similar bacteria composition was observed in the nasal samples of the control group (Fig 4A).

The SIMPER analysis predicted a total of 14 OTUs from the control group that were the primary drivers influencing the beta diversity differences between pre- and post-surgery nasal communities (S5 Table). The impact of general anesthesia on relative abundance of each genus is shown in (Fig 2C; S5 Table). Of the 14 SIMPER-detected OTUs from control group comparisons, OTU1 had the highest contribution to differences between pre- and post-surgery samples and was classified to the genus *Staphylococcus*. OTU1 in the control group comprised on

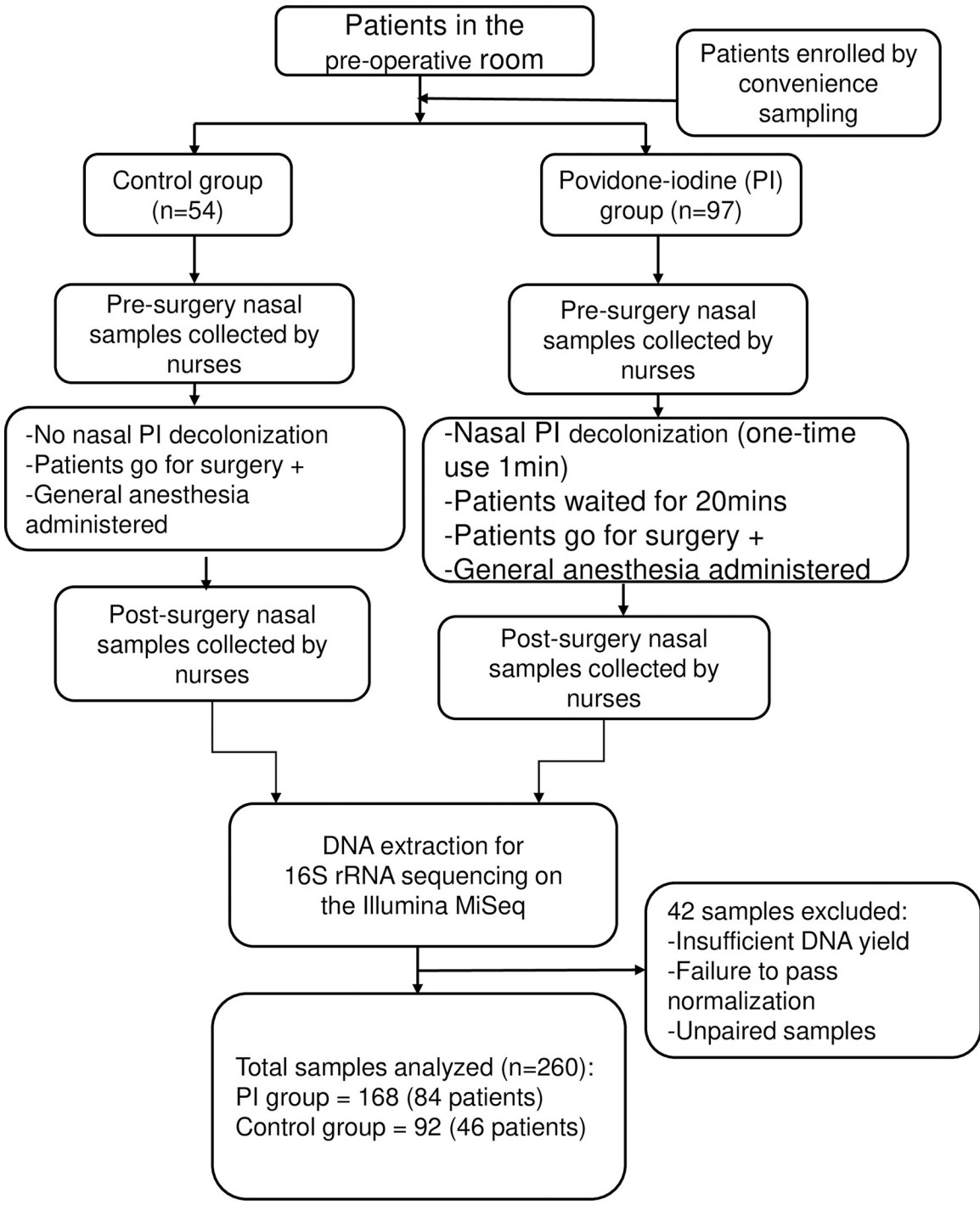

**Fig 1. Study design components.** A quality improvement study schematic outlining sample collection, nasal povidone-iodine (PI) application, and sequencing method.

average 21.65% (standard error (SE) = 0.06) of the reads in the pre-surgery control samples and 18.79% (SE = 0.06) in the post-surgery control samples, representing a 13.2% decrease. However, this difference was not statistically significant (FDR adjusted P = 0.991). In the control group, OTU2, classified to the genus *Corynebacterium*, and its abundance increased from

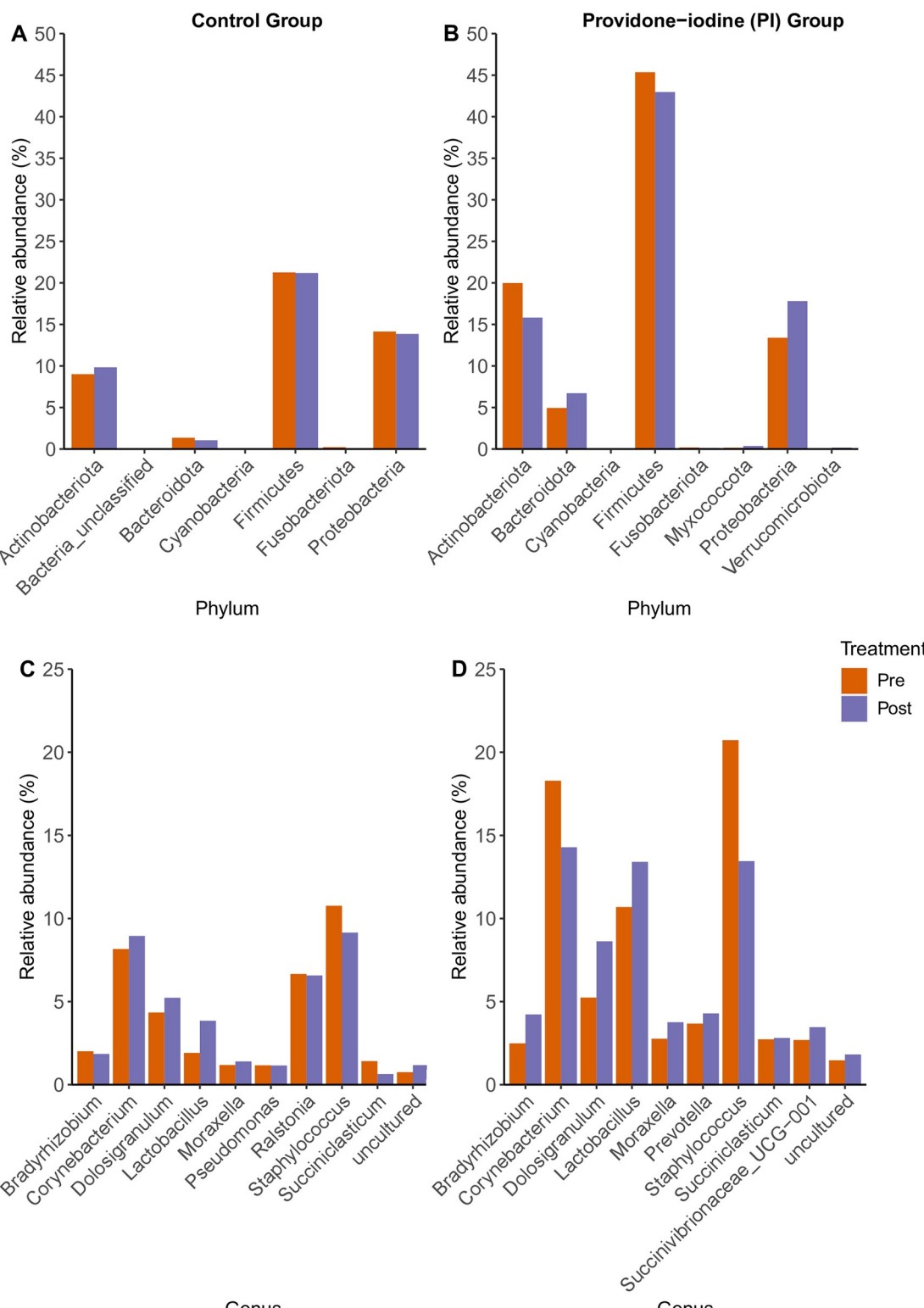

**Fig 2. Taxa composition with relative abundance.** Bar plots display relative abundance (%) of top 10 phyla and genera composition between pre- and post-surgery nasal microbiota in the control group (A and C) and PI group (B and D). Rare taxa are classified as "uncultured". The legend displays the color coding of genera and phyla to which these taxa belong.

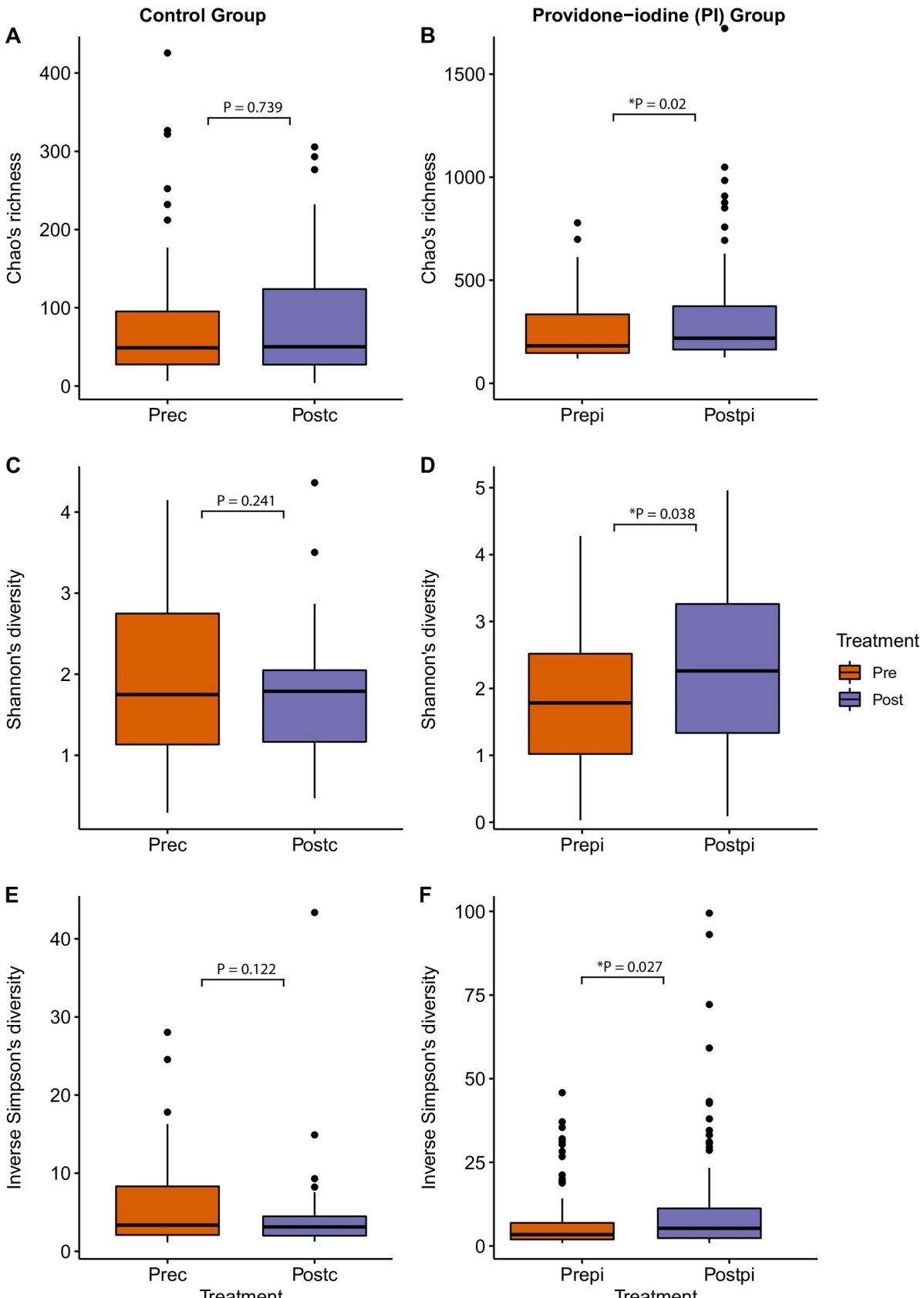

**Fig 3. Alpha diversity of bacteria community.** Box and whisker plots to compare bacteria community alpha diversity between pre-surgery and post-surgery in control (Prec, and Postc) and povidone-iodine (pi) (Prepi and Postpi) groups. (A) and (B) Chao's richness, (C) and (D) Shannon's diversity, and (E) and (F) Inverse Simpson's diversity. P-value was based on pairwise sample comparison in Wilcoxon signed-rank test. Significant values (p-values) between pre- and post-surgery are displayed on the plots. P-values <0.05 are considered significant.

16.48% (SE = 0.05) pre-surgery to 18.29% (SE = 0.06) post-surgery, representing a 11.0% increase. Similarly, this difference was not significant (FDR adjusted P = 0.944) S5 Table.

The other SIMPER-detected OTUs that explained differences between pre- and post-surgery samples in control group included bacteria classified to the genera *Ralstonia, Dolosigranulum, Lactobacillus, Bradyrhizobium, Moraxella, Lawsonella, Streptococcus, Janthinobacterium, Neisseriaceae;uncultured, Pseudomonas, Succiniclasticum,* and *Succinivibrionaceae_UCG-001* (S5 Table). None of these were found to be differentially abundant after correcting for multiple testing (P > 0.05) (S5 Table).

**Response of nasal bacterial diversity to PI.** In the PI group, we observed diverse bacterial communities with varied community compositions (Fig 2B and 2D and S6 Table). Some of the phyla included Actinobacteriota, Firmicutes, Proteobacteria, and Fusobacteria. We also observed bacteria in the genera *Staphylococcus, Moraxella, Corynebacterium* and *Lactobacillus.*

In the PI group, changes in bacterial diversity between pre- and post-surgery nasal samples were observed (Fig 3B, 3D and 3F). All three alpha diversity metrics—Chao's richness estimate (P = 0.02), Shannon's Diversity Index (P = 0.038) and Inverse Simpson index (P = 0.027)–were increased in the post-surgery patient samples, relative to pre-surgery (Fig 3B, 3D and 3F).

Comparison of the Bray-Curtis dissimilarity showed that the overall bacterial community composition between pre- and post-samples were different (P = 0.024, PERMANOVA, nMDS stress = 0.20) (Fig 4B). A similar species composition was also observed in the nasal samples in the PI group (Fig 4B).

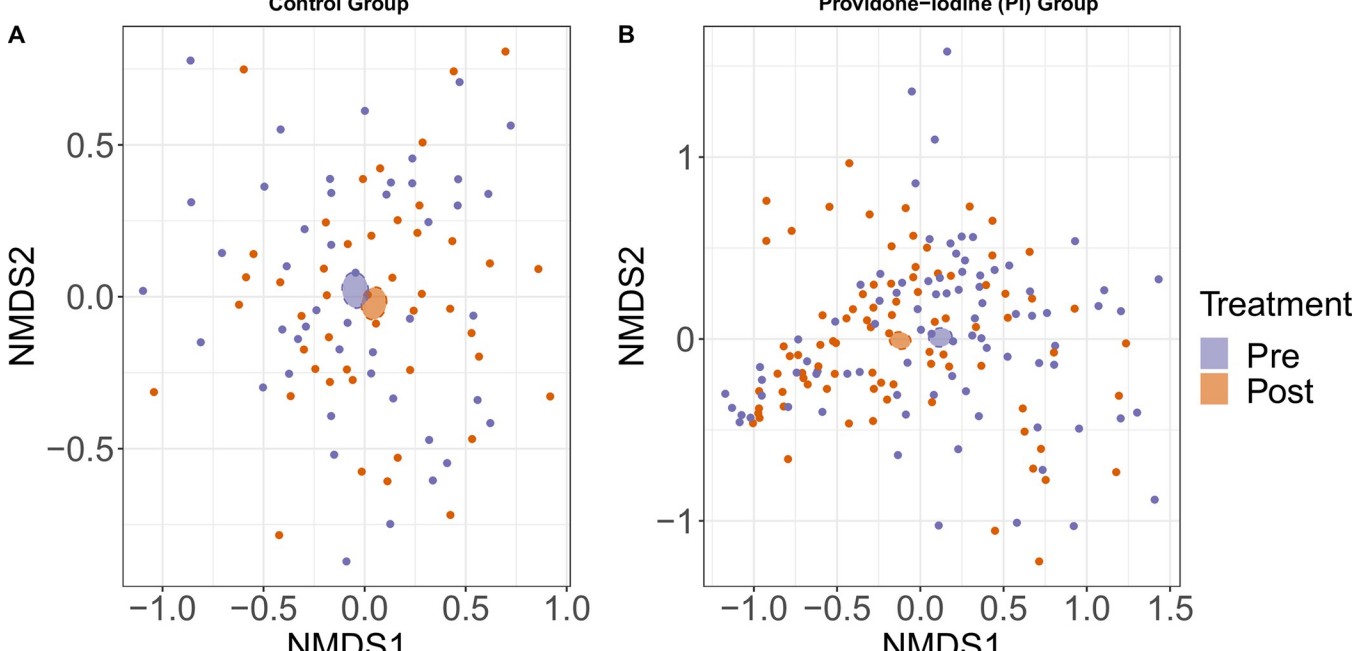

**Fig 4. Beta diversity of bacterial community composition.** Differences of nasal bacterial community compositions across treatments groups. Non-metric multidimensional scaling (nMDS) plot of the Bray-Curtis dissimilarity for bacterial community of nasal samples collected pre- and post-surgery from (A) control group and (B) povidone-iodine group. Ellipses illustrates standard error around centroid of pre- and post- nasal community samples in each treatment group and colored by treatment. P- value <0.05 was considered a significant difference between bacterial community composition between pre- and post-surgery samples. The nMDS stress levels for PI group = 0.20 and control group = 0.25.

Our SIMPER analysis predicted 10 OTUs from PI group to be primary drivers influencing the differences between pre- and post-surgery nasal community (S6 Table). The impact of PI on the relative abundance of each genus is shown in (Fig 2B and 2D and S6 Table). In the PI group's differential abundance analysis, OTU1, which classified to the genus *Staphylococcus*, had the greatest contribution to the differences observed between pre- and post-surgery samples (Fig 2D, S6 Table). The mean relative abundance of *Staphylococcus* decreased from 22.98% (SE = 0.07) pre-surgery to 13.77% (SE = 0.04) post-surgery, representing 40.1% decrease. This trended towards being significantly reduced in the PI post-surgery group (FDR adjusted P = 0.06, S3 Table).

OTU2, classified to the genus *Corynebacterium*, was found in lower abundance (20.1% (SE = 0.06)) pre-surgery compared to post-surgery (14.54% (SE = 0.05)), representing 27.75% decrease in PI group. However, this difference was not statistically significant (FDR adjusted P = 0.197, S6 Table).

Other OTUs classified to the genera *Lactobacillus*, *Dolosigranulum*, *Moraxella*, *Bradyrhizobium*, *Succinivibrionaceae_UCG-001*, *Lawsonella*, *Prevotella*, *Succiniclasticum* were shown to has contributed to observed community-level differences between pre- and post-surgery nasal samples in the PI group (S6 Table). Of these, insufficient evidence of abundant change was observed between pre- and post-surgery samples (FDR adjusted P > 0.05, S6 Table).

**Exploratory analysis between control and PI groups.** The alpha diversity exploratory analysis showed significant difference in diversity between control-post and PI-post for Chao's richness estimate (P = 0.019) and Shannon's diversity index (P = 0.047), however the Inverse Simpson index did not show a significant difference (P = 0.113). As expected, there was no significant difference between diversity in the 46 pre-surgery samples of the control group and the randomly selected subset of 46 pre-surgery of PI samples: (Chao's richness estimate P = 0.67, Shannon's diversity Index P = 0.821, and Inverse Simpson index P = 0.561).

**Impact of anesthesia and PI on *Staphylococcus* relative abundance.** In the exploratory analysis to further determine the impact of anesthesia and PI on *Staphylococcus* relative abundance, we compared 46 control post-surgery with 46 post-surgery subsampled PI group. The PI group post-surgery had 12.59% (SE = 0.04) *Staphylococcus*, which was 33.16% lower than the control post-surgery group's relative abundance of 18.79% (SE = 0.06). The difference was insufficient to show significant difference (FDR adjusted P = 0.165). The abundance changes between 46 control group pre-surgery (21.65% (SE = 0.06)) and 46 PI group pre-surgery (25.53% (SE = 0.08)) increased by 17%. This difference was insufficient to show a significant difference (FDR adjusted P = 0.958). For quality control assessment, the phyla and genera (S1 Fig) observed were comparable to the positive control, as expected [45].

## Discussion

In this QI study, we investigated the relationship between general anesthesia and preoperative intranasal PI decolonization on nasal microbiota diversity using 16S rRNA gene amplification sequencing analysis. To the best of our knowledge, this QI study is the first of its kind. Research in this area has primarily focused on the effect of PI on SSI rates but not specifically on the nasal microbiota. The relationship between microbial diversity and infection risk has been well studied. Reports show that patients with greater microbial diversity are more resistant to infection compared to those with lower microbial diversity [46,47].

### Response of nasal microbiota to general anesthesia

In this study, all patients received general anesthesia. However, the type of anesthesia administered varied from patient to patient due to the difference in medical history and the procedure

required to improve the quality of life of the patient. General anesthesia was administered by either inhalation via mask or endotracheal tube, through an intravenous line (IV), or a combination of these. In this group, we identified both pathogen-containing and non-pathogen-containing genera with high relative abundance in the anterior nares of patients pre- and post-surgery. The nasal bacterial community composition was similar to the nasal microbiota of previous studies [48–50].

Data from the control group demonstrated that general anesthesia did not significantly alter the diversity of the nasal bacterial community. Additionally, the pre-surgery abundance of each genus was not significantly different from post-surgery samples. This suggests that the type of anesthesia administered to patients in the academic center did not significantly affect a patients' nasal microbiota at the time samples were taken. Therefore, it may not severely deplete the nasal microbiota diversity to increase risk of opportunistic pathogens spreading to other sites of the body to potentially cause infections. Although anesthesia did not exhibit significant change in nasal microbiota diversity, the presence of genera with known pathogenic species in the control group highlights the need for decolonizing pre-surgery patients with intranasal PI prior to surgery, as recommended [14]. The control group also serves as a baseline to assess if the changes in diversity that occurred after PI application was influenced by anesthesia or not.

## Response of nasal microbiota to PI

Unlike the control group, an increased bacterial diversity was detected among the PI group at the end of surgery. Further evidence depicts that the PI post-surgery compared to control post-surgery also yielded a similar outcome. These suggest the activity of PI against bacteria varied and PI may play a vital role in promoting the increased diversity. Research shows that PI can readily penetrate the cell wall of *S. aureus* within 30 seconds to kill or inhibit growth [51]. On the other hand, commensals such as *Corynebacterium spp.* can hinder PI from penetrating the cell wall due to the high mycolic acid content [52]. This may support commensals such as *Corynebacterium spp.*, allowing them to recover rapidly and colonize faster than pathogens to control diversity [52]. In addition, bacteria including *Corynebacterium* species have antagonistic mechanisms against *S. aureus* proliferation [53,54]. The increased bacteria community richness and evenness observed also supported the stability of the diversity at the end of the surgery. The persistent inhibition and bactericidal activity of PI on pathogens (e.g. *Methicillin-resistant Staphylococcus aureus* (MRSA) and methicillin-susceptible *Staphylococcus aureus* (MSSA) [51,55,56] may also be a contributing factor for commensals to thrive, compete for colonization [12,53] and maintain diversity. As such, harmful invaders may be contained by commensals from spreading to other body sites [57] especially during surgery.

## Impact of PI on abundance and prevalence of bacterial taxa

The role of an individual bacterial taxa present in the nasal community is vital [54,58]. We identified specific bacterial taxa community members that may drive the differences seen between taxa communities with and without PI. While anesthesia did not significantly impact individual genera in the control group, PI impacted individual genera differently. In the PI and control groups, the genus *Staphylococcus* (phylum Firmicutes) was the most abundant bacterial taxa across PI and control samples pre-surgery. Post-surgery, PI samples showed a significant decrease in the relative abundance of *Staphylococcus*. However, there was no such decrease in this genus between the pre- and post-surgery in the control group. This demonstrates the potential of PI to reduce the relative abundances of *S. aureus*, strains of which are recognized as the main causative agent of SSI. Other species in the genus *Staphylococcus* which

do not cause serious infections are known to modulate the abundance of *S. aureus* [58]. We also observed that the genus *Corynebacterium* (phylum Actinobacteriota) was the second most abundant genus responsible for community-level differences between pre- and post-surgery nasal bacterial communities. A report showed *Corynebacterium* species play an integral role in controlling *S. aureus* by the human cell binding competition mechanism [53]. The PI penetrability of bacteria cell walls may also play an important role [52].

PI has broad antimicrobial spectrum activity and its impact on the relative abundance of the genus *Corynebacterium*, and other commensal genera (*Prevotella*, *Lactobaccilus*, *Dolosigranulum*) in the PI group, varied and was insufficient to detect significant differences. In addition to the antimicrobial activity of PI, the high abundance of multiple commensals; *Corynebacterium*, *Prevotella*, *Lactobaccilus*, *Dolosigranulum* may contribute to the striking decrease in abundance of the genus *Staphylococcus* after PI application. These findings support previous studies that *Corynebacterium* [53], *Lactobaccilus* [59] *and Dolosigranulum* [60] negatively affect *S. aureus*.

This study had several limitations. The samples in this QI study were collected from a single academic center, not randomized, not adequately powered, and therefore, may not be generalizable. The QI study did not follow patients to determine if the nasal microbial diversity change by PI led to a reduction in rate of SSI. Therefore, it was impossible to directly relate changes in the nasal microbiota to the rate of SSI among patients. Since QI study does not allow for the collection of patient identifiers, we were unable to collect any patient-level and provider-level confounding variables such as prior MRSA infections or colonization, extra nasal colonization, co-morbidities, type of surgery, duration of surgery, age, gender, antibiotics, ethnicity and diet. We were unable to connect the type of anesthesia administered to each patient's nasal sample. Therefore, could not stratify the general anesthesia administration mode into mask via nose or mouth, tube, or others.

## Conclusions

Our data showed that intranasal PI decolonization prior to surgery increases nasal bacterial diversity among surgery patients. Therefore, preoperative intranasal PI may serve as a potential option to prevent SSI by nasal bacterial community modulation. Additionally, general anesthesia administration did not significantly alter the nasal bacterial community, evidence that it has no potential impact on the risk of escalating opportunistic pathogens from a patient's nares to other body sites to cause infections. This study was conducted in one ambulatory surgical setting with patients having limited variety of surgical procedures, therefore further studies may be required to fully understand the impact of PI decolonization nasal microbial communities.

## Supporting information

**S1 Fig. Microbial community standard composition.** Stacked bar plots of the absolute abundance of top 10 phyla and genera composition in positive control (ZymoBIOMICS® Gut Microbiome). Rare taxa are classified as "uncultured". The legend displays the color coding of genera and phyla to which these taxa belong.
(TIF)

**S1 Table. A table of samples, their coverage, read counts, alpha diversity metrics, and metadata.**
(XLSX)

**S2 Table. A table of samples and their operational taxonomy units (OTUs) in the control group.**
(XLSX)

**S3 Table. A table of samples and their operational taxonomy units (OTUs) in the PI group.**
(XLSX)

**S4 Table. The relative abundance of the top 10 phyla and genera.** The relative abundance of the top 10 phyla and genera found in pre-surgery and post-surgery samples in control and povidone-iodine groups.
(XLSX)

**S5 Table. The SIMPER analysis for the control group.** The predicted OTUs by SIMPER analysis drive differences found between pre- and post-surgery in control group.
(XLSX)

**S6 Table. The SIMPER analysis for the PI group.** The predicted OTUs by SIMPER analysis to drive differences found between pre- and post-surgery in PI group.
(XLSX)

**S7 Table. List of OTUs and their taxonomy.**
(XLSX)

## Acknowledgments

The authors would like to thank Ms. Michele Zimbric (Division of Infectious Diseases, Department of Medicine, University of Wisconsin School of Medicine and Public Health, Madison), Dr. Evelyn Hammond (Institute of Natural Resources, University of Wisconsin Madison Division of Extension), Ms. Edna Chiang, Mr. Ibrahim Zuniga Chaves (Department of Bacteriology, University of Wisconsin-Madison), and the Suen Lab for their technical support, careful reading of this manuscript, and invaluable suggestions during the study.

## Author Contributions

**Conceptualization:** Eric N. Hammond, Nasia Safdar.

**Data curation:** Eric N. Hammond, Nathan Putman-Buehler, Lauren Watson, Jared J. Godfrey, Courtney Deblois.

**Formal analysis:** Eric N. Hammond, Ashley E. Kates, Courtney Deblois, Andrew J. Steinberger, Madison S. Cox, Joseph H. Skarlupka.

**Investigation:** Jared J. Godfrey.

**Methodology:** Eric N. Hammond.

**Project administration:** Eric N. Hammond, Ashley E. Kates, Lauren Watson, Jared J. Godfrey, Nicole Brys.

**Resources:** Nathan Putman-Buehler, Nicole Brys, Andrew J. Steinberger, Joseph H. Skarlupka, Ambar Haleem, Michael L. Bentz, Garret Suen, Nasia Safdar.

**Software:** Andrew J. Steinberger, Madison S. Cox.

**Supervision:** Ashley E. Kates, Ambar Haleem, Michael L. Bentz, Garret Suen, Nasia Safdar.

**Validation:** Nathan Putman-Buehler, Courtney Deblois, Andrew J. Steinberger, Garret Suen, Nasia Safdar.

**Visualization:** Nasia Safdar.

**Writing – original draft:** Eric N. Hammond, Ashley E. Kates.

**Writing – review & editing:** Eric N. Hammond, Ashley E. Kates, Nathan Putman-Buehler, Lauren Watson, Jared J. Godfrey, Nicole Brys, Courtney Deblois, Andrew J. Steinberger, Madison S. Cox, Joseph H. Skarlupka, Ambar Haleem, Michael L. Bentz, Garret Suen, Nasia Safdar.

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
