## [Decision Letter · Decision Letter 0]

12 Sep 2022

PONE-D-22-21286A quality improvement study on the relationship between intranasal povidone-iodine and anesthesia and the nasal microbiota of surgery patientsPLOS ONE

Dear Dr. Hammond,

Thank you for submitting your manuscript to PLOS ONE. After careful consideration, we feel that it has merit but does not fully meet PLOS ONE’s publication criteria as it currently stands. Therefore, we invite you to submit a revised version of the manuscript that addresses the points raised during the review process.

The reviewers would like to see some revisions made to your manuscript before publication. Therefore, I invite you to respond to the reviewers comments and revise your manuscript.

When you revise your manuscript please highlight the changes.

We look forward to receiving your revised manuscript.

Kind regards,

Rajesh P. Shastry

Academic Editor

PLOS ONE

Journal Requirements:

2. Please provide additional details regarding participant consent. In the ethics statement in the Methods and online submission information, please ensure that you have specified whether: 1) whether the ethics committee approved the verbal/oral consent procedure, 2) why written consent could not be obtained, and 3) how verbal/oral consent was recorded. If your study included minors, please state whether you obtained consent from parents or guardians in these cases. If the need for consent was waived by the ethics committee, please include this information.

3. Please upload a new copy of Figures 2 to 4 as the detail is not clear. Please follow the link for more information: https://blogs.plos.org/plos/2019/06/looking-good-tips-for-creating-your-plos-figures-graphics/

https://blogs.plos.org/plos/2019/06/looking-good-tips-for-creating-your-plos-figures-graphics/

Additional Editor Comments:

Please revise the manuscript as per reviewers suggestions and include the justifications for the same.

Reviewers' comments:

Reviewer's Responses to Questions

**Comments to the Author**

1. Is the manuscript technically sound, and do the data support the conclusions?

Reviewer #1: Yes

Reviewer #2: Yes

Reviewer #3: Partly

2. Has the statistical analysis been performed appropriately and rigorously? 

Reviewer #1: Yes

Reviewer #2: Yes

Reviewer #3: No

3. Have the authors made all data underlying the findings in their manuscript fully available?

Reviewer #1: Yes

Reviewer #2: Yes

Reviewer #3: Yes

4. Is the manuscript presented in an intelligible fashion and written in standard English?

Reviewer #1: Yes

Reviewer #2: Yes

Reviewer #3: Yes

5. Review Comments to the Author

Reviewer #1: The authors found that povidone-iodine decolonization of the nasal cavity during surgery altered the composition of the nasal flora compared to its absence. Specifically, we have found that povidone-iodine sterilization reduces the proportion of staphylococci and increases bacterial diversity. On the other hand, general anesthesia did not alter the composition of the nasal flora.

Staphylococci are bacteria highly associated with surgical site infections. It is conceivable that staphylococci, which exist abundantly in the nasal cavity, which is the inhalation part of general anesthesia, diffuse into the body through the respiratory system due to anesthesia, increasing the risk of infection during surgery. Our results suggest that intranasal use of povidone-iodine during anesthesia may reduce the composition of staphylococcal bacteria and reduce infections during surgery. In that sense, the present results are considered to be significant.

However, this paper leaves some unclear questions.

1. Age and gender distribution of subjects is unknown. It is necessary to examine whether these two conditions are factors for changes in the bacterial flora. It should also be investigated whether the differences in the microbiota with and without povidone-iodine use differ by age and gender.

2. The subject was on antibiotic prophylaxis during surgery, but the drug was not specified. It is conceivable that the type of antibiotics affects the compositional changes of the nasal flora after surgery. Therefore, it is necessary to clarify which antibiotics were used and to clearly demonstrate the effect of different antibiotics on postoperative changes in the composition of the bacterial flora.

3. The fourth variable region of the bacterial 16sRNA gene is targeted for the analysis of bacterial flora from the nasal cavity. Whether or not this is the most suitable method for compositional analysis of the bacterial flora in the nasal cavity should be explained in detail using various literatures and findings.

4. Are subjects with chronic or acute sinusitis, various respiratory/nasal diseases, and allergic rhinitis included in the subject exclusion criteria? Do these differ in composition of the nasal flora compared to healthy subjects in the nasal cavity? Or are the proportions of these affected individuals different in the two groups? The above should be clearly indicated. If you have not considered this point, you should explain in detail why you did not consider it, or why you did not need to consider it.

5. Overall, the figures are not clean and the letters are too small. The figure should be modified to make it clear.

Reviewer #2: Dear Authors,

Thank you for conducting this study entitled "A quality improvement study on the relationship between intranasal povidone-iodine and anesthesia and the nasal microbiota of surgery patients" for possible publication in the esteemed journal "PLOS ONE". The manuscript needs major revision because of the following comments:

Materials and methods

1. You should mention the design of the study. I see only the title "Study design and participants".

2. You should define the study groups. Because this sentence "(PI group and general anesthesia only group)." is confusing for the readers.

3. Please discuss the way of randomization.

4. More detail is needed about the inclusion and exclusion criteria.

5. Please provide the reference number and date of your IRB approval letter.

6. You didn't mention the data information regarding the age, gender, comorbidities, type of operation, etc. (these might affect the results).

7. How do you calculate the sample size?

Results

1. This sentence "Patients included those scheduled for cosmetic plastic surgery – breast reduction, breast augmentation, and tummy tuck." Belongs to the inclusion criteria in the materials and methods section.

2. The writing in Figure 1 is not clear. Besides, the other figures are blurred.

3. The Figure 1 legend should be below the figure. Please do the same for other figures.

Reviewer #3: Title of the manuscript: A quality improvement study on the relationship between intranasal povidone-iodine and anesthesia and the nasal microbiota of surgery patients

Review comments

The manuscript described nasal bacterial composition on treatment with povidine-iodine treatment in surgery patients. In my opinion, the manuscript should provide more detailed information and be improved overall. The nasal tract of humans consists of a very complex and dynamic microbial ecosystem. A series of exogenous and endogenous factors can affect the establishment and nature of the microbial composition in the tract. Such as, the age, gender, diet, habits, ethnicity, the surrounding environment and so on, they are very critical factors that affect the initial colonization and the subsequent establishment process. The author should have a very clear explanation of the above-mentioned factors.

It has novel findings about the bacterial differences between the two groups, but the data analysis doesn’t seem to be appropriate to represent the real diversity. Most of all, the number of reads and OTUs seem to be underestimated. The discussion needs to be enforced. For example, the distinguished taxa between groups and genera should be discussed more. The authors have focused majorly on the generic level but the higher taxonomic level variations and similarity have to be added. The SIMPER results have to be corroborated using other statistical approaches like ASCA, Lefse or ANCOM

Specific comments

1. The metadata of the patients and their biochemical profile is missing in the MS which is a big drawback of the study. The gender/ethnicity/diet or the overall health of the individual is known the drive the variations in microbiome and it needs to be provided for the benefit of the scientific community and readers.

2. Why a V4 region was preferred over V3-V4 or V4-V5 region

3. The total number of OTU's are not provided in the results section

4. Why rarefaction curves for the groups has not been provided?

5. The sample sequencing shows too much difference. There is a wide disparity between the reads. Some sample in treated group has reads of 418, while the max ranges upto 51400. In the control group, the same ranges from 440 to 1,47,000. What is the reason for this much variance?

6. Authors have used the metaMDS package of vegan to plot the NMDS plot. The ellipse color can be transparent allowing to see the grouping of sample points.

7. Figure 2A-D can be made better by converting the same in to a stacked barplot which will help the reader. The y axis of the images is in different scales which is misleading the reader.

8. There is no data on the OTU classification and their actual numbers in the samples as a OTU table– This data needs to be added as a supplementary.

9. What was the stress value for nMDS analysis? That should be provided in the manuscript

10. PERMANOVA results are confusing. The lines 334-335 indicates that there is difference between the PI and control groups – this could be due to the diverse patient profile which could be the reason for this difference.

Lines 88-89 “On the day of surgery, patients are also administered with an antibiotic prophylaxis”. What antibiotics were administered to the patients and whether this was the same for all the patients?

Was the antibiotic history of the patients collected as this can affect the overall outcome of the study?

Lines 254-256, when mentioning the groups, provide relative abundance in the parenthesis.

6. PLOS authors have the option to publish the peer review history of their article (what does this mean?). If published, this will include your full peer review and any attached files.

Reviewer #1: No

Reviewer #2: No

Reviewer #3: No

---

## [Author Response · Author response to Decision Letter 0]

31 Oct 2022

Additional requirements: 

2. Please provide additional details regarding participant consent. In the ethics statement in the Methods and online submission information, please ensure that you have specified whether: 

i) whether the ethics committee approved the verbal/oral consent procedure,

Authors’ reply: The University of Wisconsin Health Sciences Institutional Review Board exempted this study as a quality improvement project and therefore did not request a verbal consent statement for approval. We have indicated this on Page Lines 107-115. 

ii) why written consent could not be obtained, and 

Authors’ reply: As noted above, we did not obtain written consent because the University of Wisconsin Health Sciences Institutional Review Board exempted this study as a quality improvement project. Moreover, a written consent is not mandatory in the quality improvement project. 

iii) how verbal/oral consent was recorded. If your study included minors, please state whether you obtained consent from parents or guardians in these cases. If the need for consent was waived by the ethics committee, please include this information.

Authors’ reply: Although the nurse obtained verbal consent it was not recorded. Minors were not involved in this study. 

3. Please upload a new copy of Figures 2 to 4 as the detail is not clear.

Authors’ reply: We have corrected these figures to be clearer and have uploaded the revisions

Reviewer #1 comments

Reviewer #1 comment 1: Age and gender distribution of subjects is unknown. It is necessary to examine whether these two conditions are factors for changes in the bacterial flora. It should also be investigated whether the differences in the microbiota with and without povidone-iodine use differ by age and gender .

Authors’ reply: While we agree with the reviewer, this study was considered a quality improvement project. Because quality improvement projects are not human subjects research, we were unable to collect any patient identifiers (such as age and gender) as indicated in lines 111-112. We have also stated in lines 422-429 that “Since QI study does not allow for the collection of patient identifiers, we were unable to collect patient-level and provider-level confounding variables such as prior MRSA infections or colonization, extra nasal colonization, co-morbidities, type of surgery, and duration of surgery”. To clarify potential confounding variables, we have added age, gender, antibiotics, ethnicity, and diet (Page 18, Lines 425-426). Based on these limitations we were unable to stratify our data by age or gender or ethnicity or antibiotic use.

Reviewer #1 comment 2: The subject was on antibiotic prophylaxis during surgery, but the drug was not specified. It is conceivable that the type of antibiotics affects the compositional changes of the nasal flora after surgery. Therefore, it is necessary to clarify which antibiotics were used and to clearly demonstrate the effect of different antibiotics on postoperative changes in the composition of the bacterial flora.

Authors’ reply: We have updated our manuscript to include: “Patients were administered with intravenous antimicrobial prophylaxis: cefazolin or clindamycin if allergic to the former. Patients with known Methicillin-resistant Staphylococcus aureus colonization had vancomycin 15mg/kg in addition to cefazolin[1].” (Page 6, Lines 119-122). We were unable to clearly demonstrate the effect of different antibiotics on postoperative changes in the composition of the bacterial flora because we were not able to collect data on which patients received which treatment (this would be considered identifiable information). 

Reviewer #1 comment 3: The fourth variable region of the bacterial 16sRNA gene is targeted for the analysis of bacterial flora from the nasal cavity. Whether or not this is the most suitable method for compositional analysis of the bacterial flora in the nasal cavity should be explained in detail using various literatures and findings.

Authors’ reply: We chose the fourth variable region (V4 region) because the V4 region is well-accepted for its maximum nucleotide heterogeneity and specificity to bacterial characterization. In addition, we were interested in comparing specific target regions such as hypervariable regions (V4) to ensure consistency and comparable regions of the 16S rRNA. However, V3-V4 or V4-V5 have different target hypervariable regions of the 16S rRNA gene. Based on these, our lab uses the V4 region to characterize bacteria in many sample types and have been published in major scientific journals (Yatsunenko et al. 2012; Eggers et al 2018). Moreover, we want to keep the data reproducible with the lab protocols and to be able to compare it with future projects. V4 region is a well-accepted region for bacterial classification and characterization in any sample. So far there is no standard protocol indicating that a specific hypervariable region should be used for a specific sample. Some investigators use V4 region (Lehtinen et al, 2018), V3-V4 region (Chen et al 2022; Gan et al 2021), V1-V2 regions (Chen et al 2019) to assess the status of nasal microbiota.

Reference:

1. Yatsunenko T, Rey FE, Manary MJ, Trehan I, Dominguez-Bello MG, Contreras M, et al. Human gut microbiome viewed across age and geography. Nature. Nature Publishing Group; 2012;486:222–7. 

2. Eggers S, Malecki KM, Peppard P, Mares J, Shirley D, Shukla SK, et al. Wisconsin microbiome study, a cross-sectional investigation of dietary fibre, microbiome composition and antibiotic-resistant organisms: rationale and methods. BMJ Open. 2018;8:e019450. 

3. Lehtinen MJ, Hibberd AA, Männikkö S, Yeung N, Kauko T, Forssten S, et al. Nasal microbiota clusters associate with inflammatory response, viral load, and symptom severity in experimental rhinovirus challenge. Sci Rep. Nature Publishing Group; 2018;8:1–12. 

4. Chen F, Gao W, Yu C, Li J, Yu F, Xia M, et al. Age-Associated Changes of Nasal Bacterial Microbiome in Patients With Chronic Rhinosinusitis. Frontiers in Cellular and Infection Microbiology [Internet]. Frontiers Media SA; 2022 [cited 2022 Oct 12];12. Available from: https://www.ncbi.nlm.nih.gov/pmc/articles/PMC8891534/

5. Gan W, Zhang H, Yang F, Liu S, Liu F, Meng J. The influence of nasal microbiome diversity and inflammatory patterns on the prognosis of nasal polyps. Sci Rep. Nature Publishing Group; 2021;11:6364. 

6. Chen C-H, Liou M-L, Lee C-Y, Chang M-C, Kuo H-Y, Chang T-H. Diversity of nasal microbiota and its interaction with surface microbiota among residents in healthcare institutes. Sci Rep. Nature Publishing Group; 2019;9:1–10. 

Reviewer #1 comment 4: Are subjects with chronic or acute sinusitis, various respiratory/nasal diseases, and allergic rhinitis included in the subject exclusion criteria? Do these differ in composition of the nasal flora compared to healthy subjects in the nasal cavity? Or are the proportions of these affected individuals different in the two groups? The above should be clearly indicated. If you have not considered this point, you should explain in detail why you did not consider it, or why you did not need to consider it.

Authors’ reply: We agree that sinusitis can alter nasal microbiota. However, this data would be considered a patient identifier and as such we were not permitted to collect it for this quality improvement study. Because of this, we could not exclude patients with such conditions except those who were allergic to povidone-iodine. As a result of this we mentioned in our limitations that “we were unable to collect patient-level and provider-level confounding variables such as prior MRSA infections or colonization, extra nasal colonization, co-morbidities, type of surgery, and duration of surgery” (page18 and 19, lines 424-429). Generally, patients who attended the academic center for surgery were considered healthy. 

Reviewer #1 comment 5: Overall, the figures are not clean and the letters are too small. The figure should be modified to make it clear..

Authors’ reply: We agree with reviewer’s comment and have corrected the figures.

Reviewer #2 comments

(Materials and methods)

Reviewer #2 comment 1. You should mention the design of the study. I see only the title "Study design and participants".

Authors’ reply: We thank the reviewer for this comment. We have inserted “We applied the quasi-experimental study design” (Page 6, Line 110). 

Reviewer #2 comment: 2. You should define the study groups. Because this sentence "(PI group and general anesthesia only group)." is confusing for the readers. 

Authors’ reply: We appreciate the reviewer’s comment. The definitions can be found on Page 6, lines 122-129. "The general anesthesia only group, were those who received only general anesthesia without PI or any other nasal decolonizing agents prior to surgery. The general anesthesia only group also served as controls and hereinafter will be referred to as the control group." 

Reviewer #2 comment 3: Please discuss the way of randomization.

Authors’ reply: This study is a quality improvement study, and we applied a convenience sample to assign patients to treatments (Page7, Lines 131-132). Therefore, we did not randomize our patients.

Reviewer #2 comment 4: More detail is needed about the inclusion and exclusion criteria.

Authors’ reply: We have moved the statement: “Patients included those scheduled for cosmetic plastic surgery – breast reduction, breast augmentation, and tummy tuck.” from the results section and inserted it in the materials and methods section (Page, 7, Lines 135 - 137). Others that were not included were considered as a limitation of our study. 

Reviewer #2 comment 5: Please provide the reference number and date of your IRB approval letter. 

Authors’ reply: We have revised the ethical approval section and inserted “(approval number 2019-0466, May 6, 2019)” (Page 6, Lines 114). 

Reviewer #2 comment 6: You didn't mention the data information regarding the age, gender, comorbidities, type of operation, etc. (these might affect the results).

Authors’ reply: We have addressed this comment in reviewer #1 comment 1 above. While we agree that addressing potentially confounding variables would strengthen our results, we were not able to collect any patient identifiable information for this study due to the designation as quality improvement by our IRB.

Reviewer #2 comment 7: How do you calculate the sample size?

Authors’ reply: Due to the nature of quality improvement studies, there is no minimum sample size we hoped to achieve. However, the research community usually considers a sample of 25-50 as sufficient to obtain effect size estimates. Moreover, it is not required to calculate power in quality improvement studies. We chose a sample size of 151 due to the feasibility and practicality of enrolling individuals to undertake such a study. The goal was to obtain effect size estimates, so formal sample size is unnecessary. We discussed this as part of the limitation of this study (Page 18, Line 421).

Results:

Reviewer #2 comment 1(Results): This sentence "Patients included those scheduled for cosmetic plastic surgery – breast reduction, breast augmentation, and tummy tuck." Belongs to the inclusion criteria in the materials and methods section.

Authors’ reply: We agree and have moved it to this materials and methods section (Page, 7, Lines 135 - 147).

Reviewer #2 comment 2(Results). The writing in Figure 1 is not clear. Besides, the other figures are blurred.

Authors’ reply: We agree with reviewer’s comment and have corrected the figures.

Reviewer #2 comment 3 (Results): The Figure 1 legend should be below the figure. Please do the same for other figures.

Authors’ reply: We followed PLOS ONE submission guidelines to separate the figures from the legend. The legend should be inserted in the text and the figure should stand alone. 

 

Reviewer #3 comments:

Reviewer #3 comment 1: The metadata of the patients and their biochemical profile is missing in the MS which is a big drawback of the study. The gender/ethnicity/diet or the overall health of the individual is known the drive the variations in microbiome and it needs to be provided for the benefit of the scientific community and readers.

Authors’ reply: Please see the response to reviewer 1 above.

Reviewer #3 comment: 2. Why a V4 region was preferred over V3-V4 or V4-V5 region

Authors’ reply: We have addressed this comment in reviewer #1 comment 3 above.

Reviewer #3 comment: 3. The total number of OTU's are not provided in the results section.

Authors’ reply: We have updated our manuscript to reflect this as follows: 

We have inserted “prior to decontamination” (Page 11, Line 243).

We delete “After clean-up in mothur, 3,061,702 high-quality reads remained (mean = 9,876.46, SD = 16,235.97).” (Page 11, line 244). We noticed the statement was redundant and was not part of the analysis. 

We inserted “Following decontamination and clean up in mothur, 2,990,013 sequences (mean = 8,398.91, SD = 14,638.16) with 14,200 OTUs remained.”(Pages 11, Lines 244-245), to provide detail information on the total raw sequences and total OTU used in our analysis.

Reviewer #3 comment 4: Why rarefaction curves for the groups has not been provided?

Authors’ reply: We did not use rarefaction curves to determine normalization cutoff point. We chose a well-accepted approach, Good’s coverage which does not required rarefaction curve to determine the best sampling depth. This approach is consistent with our lab protocol and our numerous published manuscripts in major scientific journals such as (Chiang et al. 2022; Pilch et al 2021). 

References:

Chiang E, Deblois CL, Carey HV, Suen G. Characterization of captive and wild 13-lined ground squirrel cecal microbiotas using Illumina-based sequencing. Animal Microbiome. 4: 1. 

Pilch HE, Steinberger AJ, Sockett DC, Aulik N, Suen G, Czuprynski CJ. Assessing the microbiota of recycled bedding sand on a Wisconsin dairy farm. Journal of Animal Science and Biotechnology. 12: 114. 

Reviewer #3 comment 5: The sample sequencing shows too much difference. There is a wide disparity between the reads. Some sample in treated group has reads of 418, while the max ranges up to 51400. In the control group, the same ranges from 440 to 1,47,000. What is the reason for this much variance?

Authors’ reply: There are several factors that may contribute to these differences. This includes sample to sample variation, sequencing error, amplification error and low DNA yield. During DNA equimolar pooling for sequencing, varied concentrations of DNA samples are pooled together to a specific concentration. However, small errors during DNA quantification and equimolar pooling can result in un-even sequencing depth across samples. As such, samples with high concentration yielded higher raw sequencing reads as compared to those with lower concentrations, causing the need for normalization so the data can be confidently compared.

Reviewer #3 comment 6: Authors have used the metaMDS package of vegan to plot the NMDS plot. The ellipse color can be transparent allowing to see the grouping of sample points.

Authors’ reply: We have deleted the nMDS plots without sample points and replaced them with ellipse color that are transparent to allow our readers to see the grouping of sample points (fig 4), as suggested. 

Reviewer #3 comment 7. Figure 2A-D can be made better by converting the same in to a stacked barplot which will help the reader. The y axis of the images is in different scales which is misleading the reader.

Authors’ reply: We agree that stacked bar is an effect method to visualize the estimation of individual taxa. However, we changed our stacked bar plots because all our pilot readers indicated that the stacked bar plot is confusing and difficult to read. Moreover, it is difficult to interpret and therefore cannot provide a meaningful interpretation of the plot. Following careful analyzes of the reviewer’s comment, we decided to maintain the bar plots to allow clarity, allow our readers to better read the results directly from the figures without estimating. This will also optimize their understanding and interpret the results accurately without a struggle. 

We have deleted figures 2A-D with unequal y-axis scale and replaced them with sharp and readable figures with equal scale.

Reviewer #3 comment 8. There is no data on the OTU classification and their actual numbers in the samples as a OTU table– This data needs to be added as a supplementary.

Authors’ reply: We have added OTU tables for PI and control groups and added taxonomy table to optimize clarity in the supplementary information, as suggested.

Reviewer #3 comment 9. What was the stress value for nMDS analysis? That should be provided in the manuscript

Authors’ reply: We have updated our manuscript and made the following additions:

We have inserted “, nMDS stress = 0.25” for control group (Page 12, Line 267).

We have inserted “, nMDS stress = 0.20” for PI group (Page 13, Lines 293-294).

We have inserted “The nMDS stress levels for PI group = 0.20 and control group = 0.25.” (Page 15 line 328).

Reviewer #3 comment 10. PERMANOVA results are confusing. The lines 334-335 indicates that there is difference between the PI and control groups – this could be due to the diverse patient profile which could be the reason for this difference.

Authors’ reply: We agree and decided to delete, “Comparison of the Bray-Curtis dissimilarity showed that the overall bacterial community composition between control and PI groups samples were different (P < 0.001, PERMANOVA).” The following additional changes were also made:

We have deleted “figure 4C”

We have deleted “(C) combined (A) and (B)”. 

We have deleted “and 4C” (Page 13, Line 312)

Reviewer #3 comment 10a. Lines 88-89 “On the day of surgery, patients are also administered with an antibiotic prophylaxis”. What antibiotics were administered to the patients and whether this was the same for all the patients?

Authors’ reply: We have addressed this comment in the reviewer #1 comment 2 above. The antibiotics prophylaxis administered to PI group was not different from that of control group. We were unable to link the type of antibiotics administered to patients.

Reviewer #3 comment 10b. Was the antibiotic history of the patients collected as this can affect the overall outcome of the study ?

Authors’ reply: Because this was a quality improvement study, we were not able to collect which patients received which antibiotic(s) and were not able to conduct any analyses based on antibiotic history.

Reviewer #3 comment 10c. Lines 254-256, when mentioning the groups, provide relative abundance in the parenthesis.

Authors’ reply: On line 254-257 we only mentioned a few of the taxa shown in figs 2A and 2B. After we carefully analyzed the reviewer’s comment, we noticed that adding individual taxon’s pre and post relative abundance may create duplication. To avoid this, instead of adding the relative abundance of each in in-text, we have calculated all the relative abundance and standard error of each phylum and genus in each group in S4 line 257. This also applies to lines 285-287 in addition to fig 2C and 2D which the reviewer did not mention. 

We inserted “See S4 Table for relative abundance data.” (Page 12, Line 257). We have also directed our readers to the location of the relative abundance of each taxon.

---

## [Decision Letter · Decision Letter 1]

22 Nov 2022

A quality improvement study on the relationship between intranasal povidone-iodine and anesthesia and the nasal microbiota of surgery patients

PONE-D-22-21286R1

Dear Dr. Hammond,

We’re pleased to inform you that your manuscript has been judged scientifically suitable for publication and will be formally accepted for publication once it meets all outstanding technical requirements.

Kind regards,

Rajesh P. Shastry

Academic Editor

PLOS ONE

Additional Editor Comments (optional):

Reviewers' comments:

Reviewer's Responses to Questions

**Comments to the Author**

1. If the authors have adequately addressed your comments raised in a previous round of review and you feel that this manuscript is now acceptable for publication, you may indicate that here to bypass the “Comments to the Author” section, enter your conflict of interest statement in the “Confidential to Editor” section, and submit your "Accept" recommendation.

Reviewer #1: All comments have been addressed

Reviewer #3: All comments have been addressed

2. Is the manuscript technically sound, and do the data support the conclusions?

Reviewer #1: Yes

Reviewer #3: Yes

3. Has the statistical analysis been performed appropriately and rigorously? 

Reviewer #1: Yes

Reviewer #3: Yes

4. Have the authors made all data underlying the findings in their manuscript fully available?

Reviewer #1: Yes

Reviewer #3: Yes

5. Is the manuscript presented in an intelligible fashion and written in standard English?

Reviewer #1: Yes

Reviewer #3: Yes

6. Review Comments to the Author

Reviewer #1: The authors responded appropriately to comments and criticisms and revised the manuscript appropriately.

Reviewer #3: All the questions raised by me have been addressed satisfactorily.

I approve the acceptance of this Manuscript.

7. PLOS authors have the option to publish the peer review history of their article (what does this mean?). If published, this will include your full peer review and any attached files.

Reviewer #1: No

Reviewer #3: **Yes: **Sudeep D Ghate

---

## [Editor Report · Acceptance letter]

28 Nov 2022

PONE-D-22-21286R1 

A quality improvement study on the relationship between intranasal povidone-iodine and anesthesia and the nasal microbiota of surgery patients 

Dear Dr. Safdar:

I'm pleased to inform you that your manuscript has been deemed suitable for publication in PLOS ONE. Congratulations! Your manuscript is now with our production department. 

Kind regards, 

on behalf of

Dr. Rajesh P. Shastry 

Academic Editor

PLOS ONE